# Entropy Alternatives for Equilibrium and Out-of-Equilibrium Systems

**DOI:** 10.3390/e27070689

**Published:** 2025-06-27

**Authors:** Eugenio E. Vogel, Francisco J. Peña, Gonzalo Saravia, Patricio Vargas

**Affiliations:** 1Departamento de Ciencias Físicas, Universidad de La Frontera, Casilla 54-D, Temuco 4811230, Chile; eugenio.vogel@ufrontera.cl; 2Facultad de Ingeniería y Arquitectura, Universidad Central de Chile, Av. Sta. Isabel 1186, Santiago 8330601, Chile; 3Departamento de Física, Universidad Técnica Federico Santa María, Avenida España 01680, Valparaíso 2390123, Chile; patricio.vargas@usm.cl; 4Los Eucaliptus 1189, Temuco 4812537, Chile; gonzalo.saravia@gmail.com

**Keywords:** mutability, Shannon entropy, critical phenomena, seismic time series, Monte Carlo simulation, statistical mechanics

## Abstract

We introduce a novel entropy-related function, non-repeatability, designed to capture dynamical behaviors in complex systems. Its normalized form, mutability, has been previously applied in statistical physics as a dynamical entropy measure associated with any observable stored in a sequential file. We now extend this concept by calculating the sorted mutability for the same data file previously ordered by increasing or decreasing value. To present the scope and advantages of these quantities, we analyze two distinct systems: (a) Monte Carlo simulations of magnetic moments on a square lattice, and (b) seismic time series from the United States Geological Survey catalog. Both systems are well established in the literature, serving as robust benchmarks. Shannon entropy is employed as a reference point to assess the similarities and differences with the proposed measures. A key distinction lies in the sensitivity of non-repeatability and mutability to the temporal ordering of data, which contrasts with traditional entropy definitions. Moreover, sorted mutability reveals additional insights into the critical behavior of the systems under study.

## 1. Introduction and Background

Clausius originally introduced the concept of entropy to formalize the second law of thermodynamics. He defined entropy, S(T), as a state function whose differential, for reversible processes, satisfies $dS=\delta Q_{\mathrm{rev}}/T$, where $\delta Q_{\mathrm{rev}}$ denotes the reversible heat exchanged and *T* is the absolute temperature. This formulation establishes a direct link between heat and temperature, thereby providing a quantitative measure of irreversibility in thermodynamic processes [1,2,3,4,5,6,7,8,9,10,11]. From a statistical perspective, entropy reflects the number of microscopic configurations that are compatible with a macroscopic state. This interpretation, pioneered by Boltzmann and later refined by Gibbs, bridges microscopic dynamics with macroscopic thermodynamic behavior. Based on the energy microstates $E_{i}$ and normalized probability $p(E_{i},T)$ of occupancy of such states, the energy is obtained by adding all the weighted contributions $E\left(T\right)=\sum_{i}^{all}E_{i}\;p\left(E_{i},T\right)$. The partition function, which encapsulates the statistical properties of the system, is given by $Z\left(T\right)=\sum_{i}^{all}e^{-E_{i}/k_{B}T}$, with the Boltzmann constant $k_{B}$, which we take as 1.0, to measure energy in units of temperature.

Within the framework of Boltzmann–Gibbs statistics, the probability of occupying a microstate with energy $E_{i}$ at temperature *T* is given by $p\left(E_{i},T\right)=\frac{e^{-E_{i}/T}}{Z(T)}$, which leads to the well-known expression for the thermodynamic entropy [12,13,14,15,16,17]:(1)$$S\left(T\right)=-\sum_{i}p\left(E_{i},T\right)\;\ln p\left(E_{i},T\right).$$

This notion was later extended by Shannon [18,19,20,21,22] to quantify information in systems using a probabilistic description of their states. If the probability of the system being in the *i*-th state at time *t* is $p_{i}\left(t\right)$, the Shannon entropy is defined as(2)$$H\left(t\right)=-\sum_{i}p_{i}\left(t\right)\;\ln p_{i}\left(t\right),$$
where *t* is an independent variable characterizing the system’s evolution or structure. Unlike S(T), the probabilities $p_{i}$ in Shannon’s formulation are not necessarily derived from equilibrium statistical mechanics; rather, they may originate from empirical laws, intrinsic properties of the system, or even from direct measurements. In such cases, the probabilities can be estimated by sampling procedures. Specifically, in the present work, we obtain the probabilities $p_{i}$ as relative frequencies, calculated as the ratio between the number of occurrences of the *i*-th state (frequency $f_{i}$) and the total number of recorded observations *R*, that is, $p_{i}=f_{i}/R$.

The primary aim of this paper is to explore and compare the more recent concept of mutability as a robust measure for quantifying the structural richness (temporal variability) of time-series data. The starting point is the size of the file compressed by a data compressor like wlzip, in what is called non-repeatability. Mutability is defined as the ratio of non-repeatability to the size of the original file, thus offering a normalized entropy-like quantity derived from data compressibility. It effectively characterizes the complexity of a sequence based on how readily its information can be compressed. Intuitively, a highly regular, repetitive time series can be compressed efficiently and therefore yields a low mutability value, whereas a sequence exhibiting irregular or continually evolving patterns resists compression and thus produces a higher mutability. By accounting for the sequential order of observations, mutability serves as a form of dynamical entropy. It complements traditional entropy approaches by capturing the evolving information content of the series beyond what static, distribution-based entropy measures reveal.

We explore and compare this normalized entropy measure, known as mutability [23], with the well-established Shannon entropy. As we show, mutability contains Shannon entropy as a limiting, static case while also capturing essential information about the system’s dynamics. This measure has been previously applied to a wide range of systems, including econophysical models [24], magnetic transitions [25,26], nematic transitions [27], wind energy systems [28], hypertension datasets [29], seismic activity [30,31,32,33], and granular matter [34].

In this work, we aim to provide a more general presentation of mutability, focusing on its conceptual and quantitative relationship with Shannon entropy. Furthermore, we introduce the concept of sorted mutability, highlighting its differences from standard mutability and illustrating its application in distinct physical contexts.

Applications span both equilibrium and non-equilibrium systems, and we distinguish between artificial and natural systems. Here, we focus on two examples: (1) Monte Carlo (MC) simulations of spin systems and (2) seismic records from selected subduction zones. In most cases, the systems’ states and their associated probabilities are not fully known (except for small systems [25,26]), and we rely instead on time-series data sampled with finite precision, representing universes of varying dimensions.

We consider two representative spin systems: **(1a)** The Ising model [35,36,37,38], describing short-range interacting magnetic moments in two dimensions. This model, known for its simplicity, is widely used as a benchmark for testing methodologies. **(1b)** A dipolar interaction model [39,40,41,42,43], involving long-range interactions in two dimensions, which extends beyond the Ising framework. As we shall see, this difference leads to significant distinctions in the behavior of entropic measures.

For the case of seismicity, we analyze two highly active regions within the so-called “Ring of Fire”, the tectonically active subduction zone encircling the Pacific Ocean: **(2a)** Southern California [44], where seismic activity poses significant risks to densely populated areas; and **(2b)** a portion of the Alaskan seismic region [33,45], which has recently experienced significant earthquake events.

This article is organized as follows. Section 2 introduces the theoretical and computational framework, including the definitions of mutability, symbolic encoding, and the compressibility-based approach. Section 3 presents the physical and real-world systems analyzed—namely, the Ising model, the dipolar spin system, and seismic datasets—used as benchmarks. Section 4 discusses the results and highlights the comparative insights provided by mutability and entropy. Finally, Section 5 outlines the main conclusions.

## 2. Methodology and Definitions

We report and compare the results on Shannon entropy and mutability for the cases chosen to illustrate their use in equilibrium and non-equilibrium systems. We present both techniques separately first, then converge on how they are applied afterwards.

### 2.1. Shannon Entropy

Shannon entropy is defined in the following way:(3)$$H=-\sum_{i}p\left(i\right)\;ln\left(p_{i}\right)\;,$$
where $p_{i}$ refers to the probability of occurrence of the i−th microstate of the system and the summation is over all the microstates. The probabilities should be normalized, namely,(4)$$\sum_{i}p_{i}=1.0\;.$$

One can characterize microstates by the system’s properties, and store a temporal sequence of one in a vector file. Let $Q\left(t_{i}\right)=Q_{i}$ be the i−th entry or record in such a file, where $t_{i}$ is the time of the measurement; the index *i* enumerates such instants.

For magnetic systems, we can use the sequence of energy, magnetization, the site-order parameter, or correlation to any order of neighbors. We use the former on this occasion. The sequence is generated by a usual Monte Carlo simulation complemented by a Metropolis algorithm (outlined below) based on the Maxwell–Boltzmann statistics with probabilities(5)$$p_{j}\propto e^{-E_{j}/\left(k_{B}T\right)}\;,$$
where $E_{j}$ is the energy of the j−th microstate, *T* is the temperature, and $k_{B}$ is the Boltzmann constant. The energy sequence $E\left(t_{i}\right)=E_{i}$ is then stored in the vector file.

For the seisms, we can use their magnitude sequence (proportional to the logarithm of the estimated dissipated energy) or time interval within previously defined geodesic coordinates. We use the former. The sequence of magnitudes can be obtained from specialized catalogs.

Let *R* be the total number of records in the magnitude sequence extracted from the catalog. In this case, the probabilities are obtained from the frequencies of seisms for each magnitude *M*. Let f(M) be the number of seisms with magnitude *M* among the *R* records. Then the probability of occurrence of a seism of magnitude *M* within this sample is(6)$$p_{M}=\frac{f(M)}{R}\;.$$

With these probabilities, we can proceed to calculate Shannon entropy for the cases considered here using Equation (Equation 3).

### 2.2. Mutability

Mutability is defined as the ratio between two sizes of files in bytes. Let *w* be the size of the file storing *R* consecutive observables, and $w^{*}$ the size of the compressed file obtained from the previous file after using an information recognizer based on data compression. Then the mutability is defined as(7)$$\zeta =\frac{w^{*}}{w}\;.$$
Thus, the more the value of the observable repeats itself, the more effective the compression is, yielding low values of ζ. Not all data compressors produce significant information, since for some of them, accidental recognition of digit chains can lead to inconsistent results. For this reason, we have chosen wlzip as the appropriate information recognizer to accomplish the present task.

The data compressor wlzip compresses less than other compressors available on the web; however, it is specifically designed to recognize numerical information for specified digits, reflecting the significant numbers in the given data. The name wlzip originates from *word length zipper* [46].

To illustrate the way wlzip works, we have prepared Table 1. Although we could have considered any observable to develop this example, we chose to use a subset of 32 earthquakes from the California dataset that occurred close to the M 5.4 event on 5 June 2021. The magnitudes (rounded to one decimal place as most catalogs do) are listed in Table 1. The first column, labeled *i*, gives the sequential order (at times called natural time); the second column lists the corresponding magnitudes *M* of the 32 seisms; The third column presents the “map” of the second column prepared by wlzip.

The way wlzip constructs the third column is the following: we scan the second column from top to bottom, and each time a **new value** is found, we open a new row in the third column. Then, we write immediately to the right the distance (in number of places) of this value from the first record (origin) in column 2. Thus, we begin with the value 1.7, followed by 0 (distance to itself). Go to the second record in column 2, namely, 2.7; we write this in the second row of the third column, followed by the distance 1 to the origin of column 2. When a value repeats itself, after writing the distance, we put a comma and the number of repetitions. This happens precisely with 1.5, which appears twice, two positions after the origin, so we write 2,2 to the right of 1.5. We continue in this manner until i=11, when the value 1.7 appears for the second time; then we return to the location where we wrote 1.7 in the third row and write 10, representing the distance to the last time the value 1.7 appeared in column 2 (not from the origin). In this way, column 3 stores the different values found in column 2, with the relative distances between consecutive values. The lower the number of different values in column 2, the shorter column 3 is. Moreover, the more immediate repetitions are in column 2, the narrower column 3 is.

Column 4, labeled $f_{M}$, gives the frequency of the value leading this row in column 3. With $f_{M}$ we can calculate the probability of occurrence for this value according to Equation (Equation 6). Such probabilities $p_{M}$ are listed in the fifth column of Table 1. At this point, we realize that the Shannon entropy (basically given by column 5) represents a static manifestation of the mutability (basically given by column 3).

One interesting feature of mutability is that the third column allows us to entirely recover the second column, without loss of information. This feature is not possible with the information needed to calculate Shannon entropy. In other words, mutability depends on the dynamics that determine the sequence in row 2, while Shannon entropy utilizes the static distribution of frequencies only.

But we can go beyond regular mutability if, before applying wlzip, we sort the second row (ascending or descending) by the numerical values of the records. This reordering does not alter the physical meaning of the observable, but it offers a different perspective on the underlying data structure. In the sixth column, we give the sorted values of the observable in column 2. The corresponding symbolic representation using wlzip, constructed identically for column 3, is presented in the seventh column with weight $w^{†}$. The mutability of this sorted sequence (hereafter referred to as sorted mutability) is defined as(8)$$\zeta_{S}=\frac{w^{†}}{w}\;.$$
Similarly to Shannon entropy, sorted mutability does not possess dynamical features, and with its information, the original file cannot be recovered. By construction, the sorted mutability $\zeta_{S}$ provides a lower bound for mutability, thereby enhancing the identification of critical points, as shown in the results below.

The last row in Table 1 provides the values of both regular mutability and sorted mutability corresponding to the data in column 2. As can be seen, the latter is smaller than the former, despite having the same number of rows; however, sorted mutability produces a narrower map.

In principle, any data compression algorithm can be employed to analyze the informational content of a sequence. However, most widely available compressors on the web are designed to recognize digit patterns independently of their position, which may be efficient for general data compression but fails to retain the structural properties of the sequence. Early studies employed bzip2 to obtain mutability results. Nevertheless, unexpected overlaps between results of systems with different sizes [47] motivated the development of a state-oriented compressor, wlzip (“word length zipper”) [46]. Although wlzip typically compresses less efficiently than bzip2, rar, or other general-purpose compressors, it is explicitly designed to preserve state-related structures. Thus, using wlzip or similar state-aware compressors is essential for a meaningful definition of mutability.

### 2.3. Functions

For an observable of a given system, we can report four functions:*Non-repeatability V*, defined directly by the compressed weight: $V=w^{*}$.*Regular mutability* ζ, given by Equation (Equation 7). This function allows reversibility since the map generated by wlzip stores the locations of the values along the original chain of data.*Sorted mutability* $\zeta_{S}$, given by Equation (Equation 8). This function does not allow the recovery of the original data sequence.*Shannon entropy H*, given by Equation (Equation 2). This function does not allow the recovery of the original data sequence.

In recent studies, mutability has been compared with Tsallis entropy in the context of seismic data. It was found that the two measures provide complementary perspectives [48,49]. However, we do not include Tsallis entropy in the present analysis for two main reasons: (i) Simplicity of presentation, to focus on the relationship between classical Shannon entropy and the more recent mutability function; and (ii) relevance to the systems under study, since the magnetic simulations are based on the Metropolis Monte Carlo algorithm, which adheres to Boltzmann statistics. In this context, Tsallis entropy is not expected to capture the finer features of data sequences derived from these simulations.

## 3. Systems Under Study and Theoretical Basis

We begin by outlining the general features of the systems under study, describing their main characteristics. We consider two representative cases for each system and highlight the key differences in their respective modeling approaches.

(1)**First system: Square spin lattice.** At a given temperature *T*, the system’s energy E(T) is computed using the appropriate Hamiltonian, as detailed below. The time evolution is simulated via a Monte Carlo (MC) procedure [50,51,52,53] in which, at each time step *t*, a spin (or magnetic moment) is randomly selected and temporarily flipped. The resulting energy difference δ (defined as the energy before minus the energy after the flip) is evaluated. If δ>0, the flip is accepted unconditionally and E(t) is updated. If δ<0, the flip is accepted with a probability governed by the Metropolis criterion.This process is carried out over 20R MC steps at each temperature to ensure equilibration (with *R* defined per case). A subsequent sequence of 20R MC steps is then used to collect data: every 20 MC steps, the value of a relevant observable is recorded, yielding a total of *R* entries. The most probable value of the observable is taken as its average over these *R* measurements. The temperature is then updated as $T_{i+1}=T_{i}+\Delta T$, with ΔT=0.1, unless otherwise specified.
(1a)**Ising magnets.** In this case, the magnetic moments $S_{j}$ can take values +1 (aligned with +y) or −1 (aligned with −y). Only nearest-neighbor interactions are considered, described by the standard exchange Hamiltonian [54,55,56]:(9)$$H_{X}=-J\sum_{\langle j,k\rangle }S_{j}\cdot S_{k}\;,$$
where the summation is over all nearest-neighbor pairs, and J>0 denotes a ferromagnetic coupling constant. Indices (j) and (k) denote distinct lattice sites. Free boundary conditions are imposed, which are particularly suitable for modeling nanoscale systems.(1b)**Dipolar magnets.** In this configuration, narrow ferromagnets with strong shape anisotropy are positioned at the vertices of the same square lattice and aligned along the *y*-axis. These magnets are treated as point dipoles, assuming their physical size is much smaller than the lattice spacing. Their interactions are mediated by the demagnetizing field, modeled via dipole–dipole interactions. The Hamiltonian contribution, $\Delta H_{D,j}$, due to a dipole $S_{j}$ interacting with all other dipoles $S_{k}$, is given by [39,40,41,42,43](10)$$\begin{array}{c} \Delta H_{D,j}=\frac{\mu_{0}}{4\pi }\sum_{k(k\neq j)}\frac{S_{j}\cdot S_{k}-3\left(S_{j}\cdot {\hat{r}}_{k}\right)\left(S_{k}\cdot {\hat{r}}_{j,k}\right)}{|r_{j,k}{|}^{3}}, \end{array}$$
where $r_{j,k}$ is the vector from site (j) to (k), and ${\hat{r}}_{j,k}$ is its unit vector. The total Hamiltonian is obtained by summing over contributions, avoiding double counting. This is equivalent to summing over all dipole pairs. The notation $S_{j}$ is retained for consistency with the exchange model.

As shown in Figure 1, the same lattice geometry is used for both the Ising and dipolar systems. In both cases, the total number of magnetic configurations is $2^{N}$, though the pathways by which these states are accessed—e.g., by increasing temperature—differ markedly.

An essential advantage of the information-theoretic approach is that it identifies states based on matching significant digits of numerical observables. Since only relative values are needed, we adopt J=1 for the Ising case and set $\mu_{0}/4\pi =1$ and the lattice spacing to unity in the dipolar case. This simplifies the simulations while preserving the relevant dynamical structure of the system. We present the Monte Carlo (MC) simulation results exclusively for the cases of pure exchange and pure dipolar interactions. Under the above conditions, the transition temperatures are comparable in both cases [40].

(2)**Second system: Seismic activity.** The United States Geological Survey (USGS) provides a comprehensive catalog of seismic events in the United States. From this database, we extract a time series of earthquake magnitudes exceeding a given threshold (typically M 1.5) within a predefined geographical “rectangle” and depth range. Two specific regions are considered:
(2a)**California seismicity.** Data is selected from the region bounded by longitudes 115° W to 119° W and latitudes 31° N to 35° N, with depths limited to 35 km. The data extraction spans the period from 1 January 1994 to 31 December 2023, resulting in a total of 131,459 seismic events. This region includes the M 7.2 *El Mayor–Cucapah earthquake* of 4 April 2010.(2b)**Alaska seismicity.** Data is extracted from a smaller region defined by longitudes 157.3° W to 158.2° W and latitudes 54.5° N to 55.5° N, down to depths of 70 km. The time span from 1 January 2020 to 31 December 2023 includes 629 recorded earthquakes. This region is notable for recent intense seismic activity, including the M 8.2 Chignik earthquake of 29 July 2021. Due to the relatively low number of events, this dataset presents an opportunity to test information-theoretic methods under sparse data conditions.

## 4. Results and Discussion

Figure 2 presents the non-repeatability results for a data sequence corresponding to a spin lattice with exchange interactions. We consider $R=1.2\times 10^{5}$ records. As the observation window increases (i.e., larger *R*), the differences between results for different lattice sizes become more evident. However, obtaining precise proportions would require not only larger samples but also larger lattice sizes (to minimize boundary effects) and a broader temperature range, which lies beyond the scope of the present article.

Figure 3 displays both regular mutability (upper curves) and sorted mutability (lower curves) for Ising systems with L=64 and L=128. The regular mutability exhibits the typical shape of an entropy curve as a function of temperature: starting at zero, increasing gently, then more sharply, and finally saturating. Notably, the curves present a single maximum around T≈2.27, which corresponds to the well-known critical temperature of the two-dimensional Ising model [36].

The sorted mutability reveals a single, well-defined peak at the critical temperature, suggesting it is a particularly effective indicator of critical behavior. Furthermore, the sharpness of this peak increases with system size, reinforcing its usefulness.

There is an even more important difference between regular and sorted mutability: the former does not collapse with size, while the latter collapses. Let us define the collapsed sorted mutability as(11)$$\zeta_{SC}=\frac{\zeta_{S}}{L}\;.$$
Clearly, the same expression applies to the collapse of regular mutability but its lack of collapse is obvious from Figure 3.

The collapse of the sorted mutability can be seen in Figure 4 for three different sizes. Although the collapse is not perfect, we should remember that free boundary conditions are applied, which makes size differences for small systems.

We now turn to the second magnetic system, governed by dipolar interactions. Figure 5 shows the non-repeatability function for three lattice sizes. The data confirm the ordering $V_{L=16}\left(T\right)<V_{L=32}\left(T\right)<V_{L=64}\left(T\right)$. With sample sizes exceeding $1.2\times 10^{5}$ records and larger lattices, the tendency to reach higher values of non-repeatability is maintained. The maxima become more pronounced and shift slightly toward higher critical temperatures as the system size increases, approaching the well-known critical temperature $T_{C}=2.27$, similar to the case with exchange interactions. This agreement arises from setting the magnitude scaling factors to unity in both cases.

In the left panel of Figure 6, we compare Shannon entropy, regular mutability, and sorted mutability for L=32, using $R=4.8\times 10^{5}$ records (corresponding to $9.6\times 10^{6}$ Monte Carlo steps, with an equal number of equilibration steps). We selected the dipolar system for this comparison due to its less monotonic behavior with temperature. Although the plots use different scales, the vertical ranges have been normalized for comparability.

As noted from Table 1, Shannon entropy is intrinsically linked to mutability, and sorted mutability is derived from the same symbolic compression scheme. Consequently, all three quantities follow a similar qualitative trend: starting at zero, growing with temperature, reaching a peak at the critical point, and then decreasing toward higher temperatures. However, their specific temperature dependence reveals distinct sensitivities. In particular, sorted mutability displays the sharpest peak at the critical temperature, indicating it is the most precise among the three in identifying phase transitions.

The right panel of Figure 6 shows regular mutability for the same system and parameters but for varying observation window sizes *R*, as specified in the inset. The lowest curve in the right panel corresponds to the same regular mutability curve (in black) shown on the left. The figure demonstrates that the curves coincide at low temperatures but diverge near the critical region. As *R* increases, the maximum mutability value decreases and the curves flatten out at high temperatures. This behavior is expected to continue until it converges to the classical entropy curve in the thermodynamic limit: a mildly increasing curve with an inflection point and eventual saturation at high temperatures.

Therefore, the peak in mutability seen in Figure 6 is a finite-size effect due to the limited observation window. Nonetheless, this feature can be advantageous for locating critical points: by tuning the observation window *R*, one can efficiently estimate $T_{C}$ within practical computational timescales, depending on the system under investigation.

We now turn our attention to seismic activity, beginning with the Los Angeles/San Diego region in California. Figure 7 shows the Shannon entropy, mutability, and sorted mutability for overlapping time windows of r=128 consecutive events (with an overlap of one event). Here, *r* denotes the number of records used for information processing, which is significantly smaller than the values of *R* used for the magnetic system simulations. The aim is to explore whether small but significant variations can be detected in natural systems, where the number of data entries is limited—first by nature and second by the sensitivity of measurement instruments.

The vertical axes have been adjusted so that all three curves span approximately the same maximum range (about 0.65 units on the left axis, measured between the 2012 maximum and the corresponding minimum around 2000).

Several observations follow:The similarity of the three curves confirms the connection between Shannon entropy and mutability.Major earthquakes tend to coincide with upward spikes in the curves; however, the converse is not always true—some spikes are not linked to single large events, indicating the possible influence of local activity or seismic swarms.Mutability curves more clearly reveal the undulating trends in the data.Downward behavior associated with aftershock regimes is better captured by mutability curves.Mutability exhibits richer texture than Shannon entropy, with greater amplitude and clearer resolution of consecutive segments.Sorted mutability provides slightly more detail than regular mutability—for example, the downward triplet near 1995, sharper peaks around 2010, and broader oscillation ranges within the same vertical span.

The case of the Alaska earthquake of magnitude $M_{w}=8.2$ in 2021 is quite different, as seismic activity in this region is sparse. The dataset for the rectangular zone bounded by 54.5–55.5° N and 157.3–158.2° W consists of only 629 events recorded over 24 years. This case is included to test the method under extreme data scarcity, in order to verify whether the features observed in the California analysis still hold. It is known that seismic frequency decreases in the lead-up to major earthquakes, a pattern previously identified in this same region [33].

For this analysis, the geographic region was intentionally narrowed to accentuate data scarcity. Although the earthquake catalog begins in 2000, the plots in Figure 8 start in 2012, when the first set of 64 events became available for computing mutability. Shannon entropy plots were excluded from this figure, as they offer no additional information beyond what is captured by regular and sorted mutability.

The upper panel of Figure 8 presents the regular mutability for this sequence of 629 magnitudes, using overlapping windows of r=64 events. The initial mutability values are below 850. Around the end of 2016, mutability increases stepwise to approximately 900, reflecting a period of variable seismic activity. In early 2019, a drop in mutability indicates repetitive values characteristic of a quiescent phase. The mutability then oscillates until the $M_{w}=8.2$ earthquake of 2021 enters the 64-event window, producing a marked spike due to the extreme diversity of values. This is followed by a sharp drop (a downward spike), signaling the onset of the aftershock regime. After a brief recovery, another smaller dip occurs, reflecting continued energy relaxation. By the end of 2023, mutability drops to low values, due to the occurrence of several moderate earthquakes in or near the selected region, sustaining a moderate-aftershock phase. In this context, low mutability may indicate ongoing energy dissipation, potentially reducing the likelihood of an imminent large event.

The lower panel of Figure 8 shows sorted mutability computed with the same parameters. A comparison with regular mutability reveals several key points:

1. The sorted mutability values are lower than regular mutability, consistent with previous observations in magnetic systems. 2. The major features of the regular mutability curve—especially the spikes associated with the 2021 event and its aftermath—are clearly preserved. 3. General trends and curvature are more pronounced in the sorted mutability plot. 4. The texture of the sorted mutability curve is richer; notably, oscillations starting at the end of 2020 (potential precursors to the main shock?) exceed the 2017–2018 maxima.

These results confirm the utility of mutability-based measures, even in data-limited natural systems.

## 5. Conclusions

The non-repeatability function V(T) increases with system size (i.e., the number of elements in the lattice), suggesting a possible tendency toward additivity. However, this behavior is not yet fully resolved due to finite equilibration and observation times. This trend is observed in both exchange and dipolar interaction systems, though it is more pronounced in the latter. The enhanced effect for dipolar systems is likely due to their richer dynamics and the influence of long-range interactions, which allow broader sampling in configuration space.

Mutability incorporates Shannon entropy as a special case, particularly when the symbolic mapping produced by the wlzip compressor corresponds directly to the frequency of states. This relationship is confirmed by the observation that mutability curves consistently lie below the Shannon entropy curves in spin systems, enabling a clearer identification of critical points. Notably, the maxima observed in mutability near the critical temperature are artifacts arising from finite observation times.

Sorted mutability produces sharper peaks at critical points compared to regular mutability, making it a valuable tool for detecting phase transitions. This measure can be tuned by adjusting the system size and the sampling window, offering flexibility and precision in the analysis of complex systems. Sorted mutability behaves as an extensive quantity, scaling proportionally with the system size *L*. Consequently, when the sorted mutability is normalized by the system “diameter” *L*, the resulting curves collapse onto a single curve. Among the quantities analyzed in this work, sorted mutability is therefore the one that most closely resembles entropy in its scaling behavior.

In the context of seismic activity, both mutability and Shannon entropy effectively characterize temporal patterns. Upward spikes in Figure 7 correspond to either major earthquakes or clusters of moderate events (swarms), while low values are typically associated with aftershock regimes characterized by repeated low-magnitude events. Additionally, low mutability values may reflect continuous energy release via moderate events in highly active regions.

Sorted mutability provides a more refined view of criticality, both in spin systems and, to a certain extent, in seismic data. It offers enhanced texture, a better definition of temporal structures, and more pronounced contrast in identifying key dynamical features.

In summary, regular mutability—and even more so, sorted mutability—offers significant advantages over Shannon entropy. Sorted mutability reveals deeper structure in data sequences, captures oscillatory trends with enhanced clarity, and offers more sensitive detection of aftershock sequences and phase transitions.

## Figures and Tables

**Figure 1 entropy-27-00689-f001:**
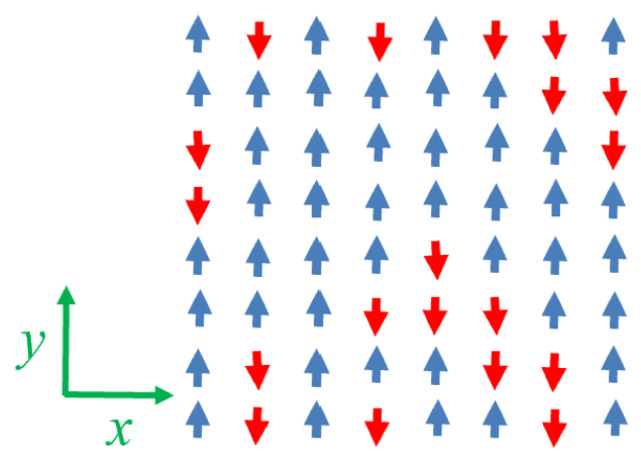
Geometry of an 8 × 8 lattice in the xy-plane. Magnetic moments can point upward or downward along the *y*-axis, corresponding to parallel or antiparallel orientations. These moments may represent Ising spins or highly anisotropic magnetic dipoles, treated as dimensionless quantities. Lattice sites are denoted by the index (j).

**Figure 2 entropy-27-00689-f002:**
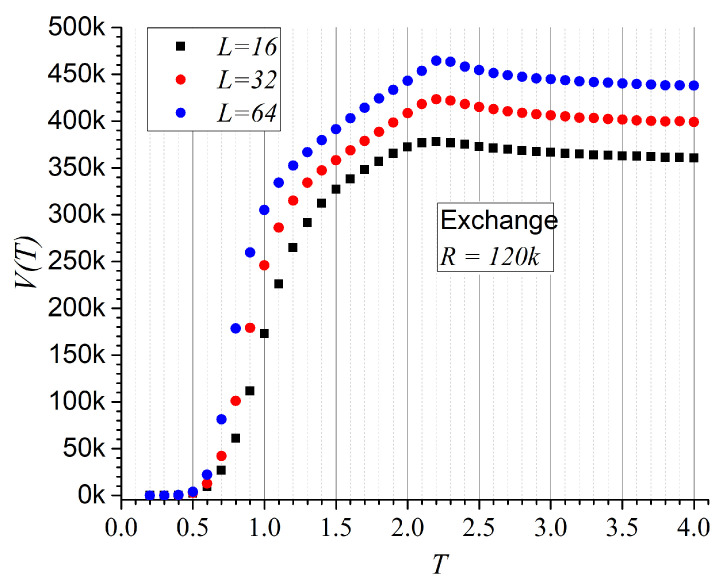
Average non-repeatability V(T) for exchange interaction only as a function of temperature *T*, for three different lattice sizes. The factor *k* corresponds to $10^{3}$.

**Figure 3 entropy-27-00689-f003:**
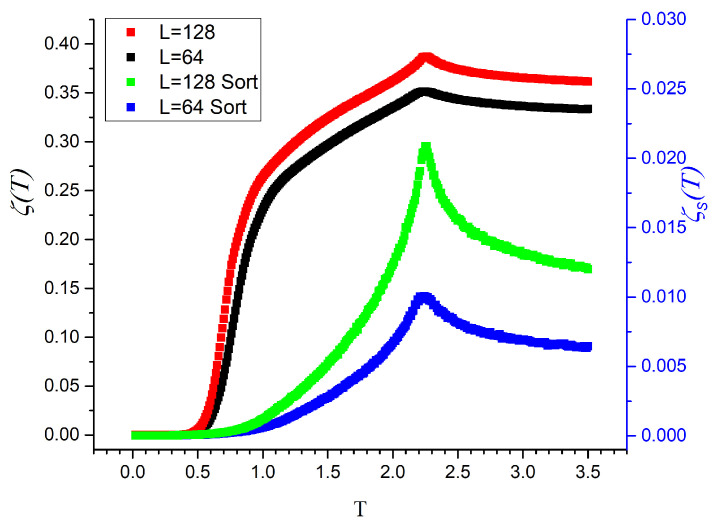
Mutability for Ising systems of lattice sizes L=64 and L=128. The two upper curves correspond to regular mutability, while the two lower curves correspond to sorted mutability for the same original data.

**Figure 4 entropy-27-00689-f004:**
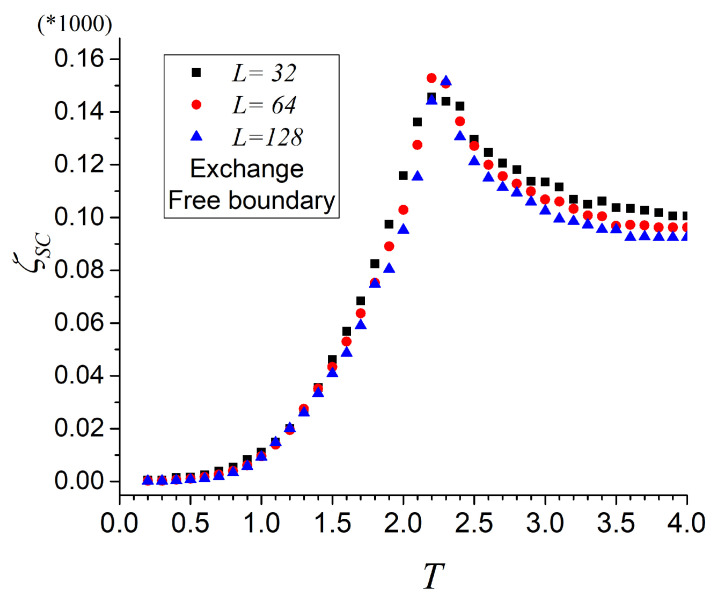
Collapsed sorted mutability for Ising systems, after Equation (Equation 11) for three lattice sizes as indicated in the inset. The actual ordinate values are amplified 1000 times.

**Figure 5 entropy-27-00689-f005:**
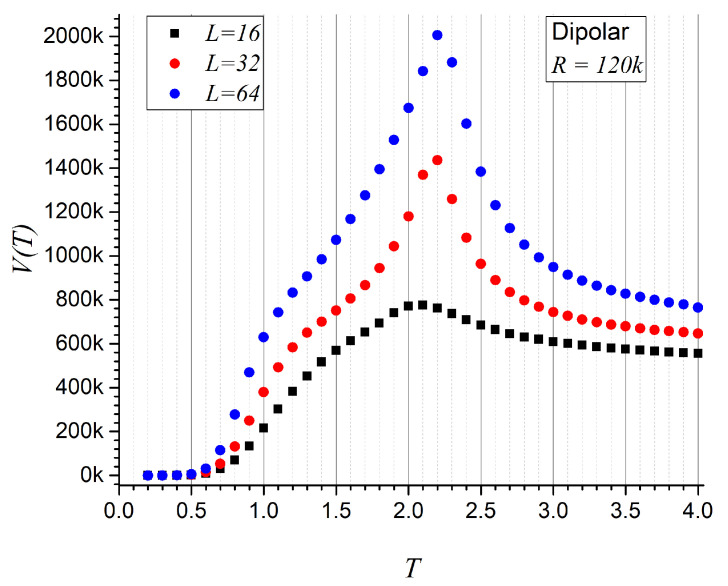
Average non-repeatability V(T) for the dipolar interaction only, as a function of temperature *T*, for three different lattice sizes. *k* indicates a factor of 1000.

**Figure 6 entropy-27-00689-f006:**
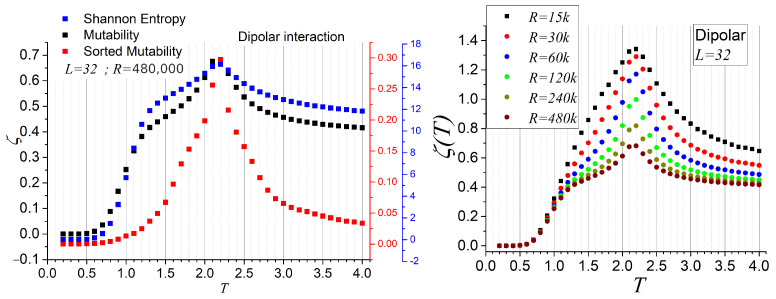
(**Left**): Shannon entropy, regular mutability, and sorted mutability for L=32, computed over 480,000 registers. (**Right**): Regular mutability for L=32 with different observation window sizes *R*, as indicated in the inset. *k* denotes a factor of 1000.

**Figure 7 entropy-27-00689-f007:**
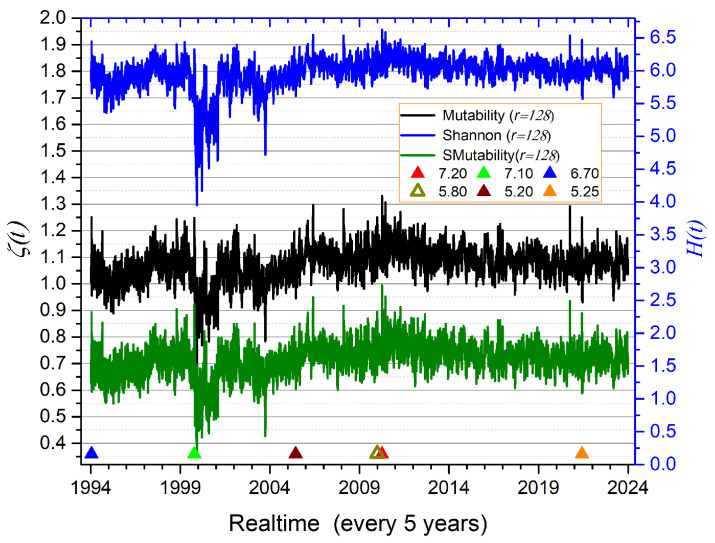
Shannon entropy, mutability, and sorted mutability for 131,459 earthquakes recorded between 1994 and 2023, with magnitudes $M_{w}\geq 1.5$, depths up to 30 km, and epicenters located within the rectangular region around Los Angeles and San Diego, as defined in the text. The occurrence times of major earthquakes are marked by symbols described in the inset.

**Figure 8 entropy-27-00689-f008:**
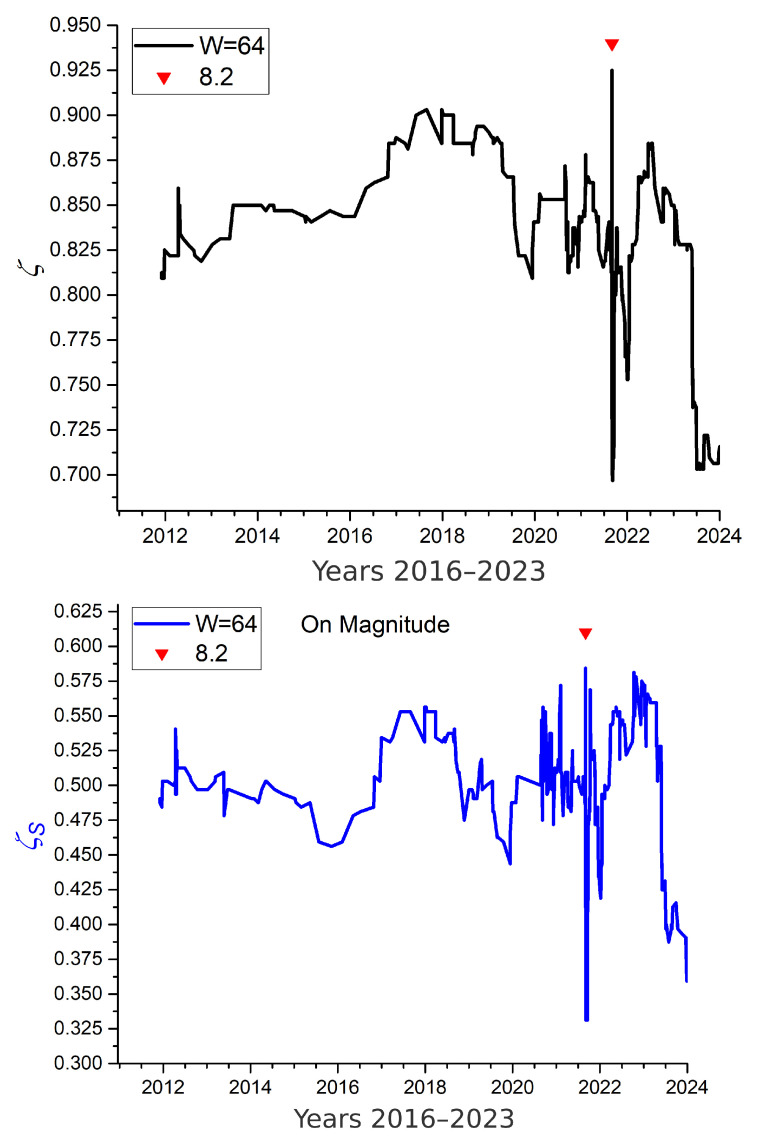
Mutability analysis of seismic data in the Alaska region. Top: Regular mutability. Bottom: Sorted mutability.

**Table 1 entropy-27-00689-t001:** Segment of a seismic dataset retrieved from the USGS, consisting of R=32 consecutive earthquake events. The columns are (1) event index *i*, (2) earthquake magnitude *M* (weight *w*), (3) compressed symbolic representation of *M* lead by each different *M* value in bold (weight $w^{*}$, using wlzip), (4) frequency f(M), (5) probability $p_{M}=f\left(M\right)/R$, (6) sorted magnitudes (weight *w*), and (7) compressed symbolic map of the sorted magnitudes (weight $w^{†}$). Mutability values ζ are shown in the last row.

*i*	*M*	Map of *M*	$f_{M}$	$p_{M}$	SM	Map of SM
1	1.7	**1.7** 0 10 12 9	4	0.12500	1.5	1.5 0,2
2	2.7	**2.7** 1	1	0.03125	1.5	1.6 2,4
3	1.5	**1.5** 2,2	2	0.06250	1.6	1.7 7,4
4	1.5	**1.8** 4	1	0.03125	1.6	1.8 11
5	1.8	**1.6** 5 2 2 4,2	5	0.15625	1.6	1.9 12,3
6	1.6	**2.4** 8 15 3	3	0.09375	1.6	2.0 15
7	1.9	**2.0** 11	1	0.03125	1.6	2.2 16,3
8	1.6	**2.2** 12 9 9	3	0.09375	1.7	2.3 19
9	2.4	**3.9** 15 2	2	0.06250	1.7	2.4 20,3
10	1.6	**5.4** 16	1	0.03125	1.7	2.6 23,2
11	1.7	**2.9** 18,2	2	0.06250	1.7	2.7 25
12	2.0	**1.9** 6 18 5	3	0.09375	1.8	2.9 26,2
13	2.2	**2.6** 20 8	2	0.06250	1.9	3.4 28
14	1.6	**3.4** 25	1	0.03125	1.9	3.9 29,2
15	1.6	**2.3** 27	1	0.03125	1.9	5.4 31
16	3.9				2.0	
17	5.4				2.2	
18	3.9				2.2	
19	2.9				2.2	
20	2.9				2.3	
21	2.6				2.4	
22	2.2				2.4	
23	1.7				2.4	
24	2.4				2.6	
25	1.9				2.6	
26	3.4				2.7	
27	2.4				2.9	
28	2.3				2.9	
29	2.6				3.4	
30	1.9				3.9	
31	2.2				3.9	
32	1.7				5.4	
$\zeta \;|\;\zeta_{S}$		0.944				0.594

## Data Availability

The original contributions presented in this study are included in the article. Further inquiries can be directed to the corresponding author.

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
