# Peer review of "Entropy Alternatives for Equilibrium and Out-of-Equilibrium Systems"

_entropy, 2025, doi:10.3390/e27070689_

Round 1

Reviewer 1 Report

Comments and Suggestions for Authors

The manuscript presents an interesting topic with potential relevance; however, in its current form, it lacks the clarity and completeness necessary for publication. The primary concern is that the core concepts are not sufficiently defined or explained within the framework of statistical mechanics, which limits the manuscript’s accessibility and scientific rigor. Many definitions and assumptions appear to rely heavily on previous publications, making the paper difficult to follow independently.

Major Comments:

  1. The introduction fails to clearly establish the statistical-mechanical context of the work. For the manuscript to stand on its own, all key concepts and quantitative definitions must be explicitly introduced. As it stands, the reader is forced to consult prior publications to interpret the terminology and methods used.
  2. Ambiguity of Terms and Methods
    • Line 141: The term “components along a DNA molecule” is vague. Is this referring to the nucleotide sequence or some other structural/functional element of DNA? Please clarify this within a statistical-mechanical or informational context.
    • Lines 144–147: Phrases like “the weight in byes” and the mention of the compressor wlzip are unclear without additional context. These terms must be explained rigorously and preferably generalized within a statistical-mechanical model. Reference 22 may contain relevant background, but this information must be explicitly summarized in this manuscript for clarity.
  3. Table 1 is particularly confusing due to inconsistent or undefined quantities:
    • The second column is labeled as $Q(t)$ (as introduced in Line 138), yet the caption refers to “earthquake magnitudes $M_w$ (weight $w$)”. Is $w = Q(t)$? If not, both quantities need clear definitions.
    • The third column, $w^*(Q,R)$, is described as the size of “a symbolic representation” or “Map of $M_w$”. These terms are not defined, nor is the method of calculation explained.
    • The presence of either one or two numbers in this column (e.g., 1.52, 2) without clarification is confusing. What do these numbers represent? Why are some rows bolded? These formatting choices should be clearly explained in either the caption or a footnote.

Recommendations:

  • Revise the introduction to define all quantities and concepts with precise language and within the context of statistical mechanics.
  • Avoid relying on previous publications for critical definitions. While references are important, they should not be essential for understanding the basic framework of this work.
  • Provide full explanations for all methods and quantities introduced in tables and figures, especially those used to represent complex or derived values.
  • Use consistent terminology throughout the manuscript, and ensure that table captions match the column labels exactly.

In summary, the manuscript has potential but is currently not suitable for publication due to significant issues with clarity and self-containment. I recommend a major revision that addresses the concerns outlined above. With substantial clarification and contextualization, particularly within the statistical-mechanical framework, the work may become suitable for publication.

Author Response

Dear Referee,

Thank you very much for your time and valuable feedback on our manuscript.
I have attached a PDF document containing our detailed responses to your comments and suggestions.

We hope that our clarifications and revisions adequately address your concerns.

Sincerely,
Francisco J. Peña, on behalf of the authors

Reviewer 2 Report

Comments and Suggestions for Authors

This manuscript explores entropy-related functions and compares them with the Shannon entropy. Two case studies to explore the effectiveness of these functions are considered: (a) Magnetic moments on a square lattice and (b) Seismic time series. The functions compared between themself and with the Shannon entropy are non-repeatability, regular mutability, and sorted mutability. The conclusion is that regular mutability and sorted mutability offer advantages over Shannon entropy, among other aspects, to detect aftershock sequences and phase transitions.

In my opinion, the manuscript is not clear and requires significant improvement before it can be considered for publication. Below, you can find some comments that can help you accomplish that. Most of them are related to the presentation of the results and the novelty of the work.

The abstract states that the manuscript aims to explore a new function known as non-repeatability and that its normalised form is referred to as mutability. But then, in the introduction section, nothing is said about the non-repeatability function but about what is called sorted mutability. This way of presenting is very confusing and makes a not good impression when starting to read the manuscript. This point must be addressed.

Presenting the systems considered without first introducing the method or framework to be considered makes it very difficult to understand the real goal of this work. I think there should be a section or something similar that explains the framework and new methodology first and then discusses the systems chosen as benchmarks.

On page 5, line 152, it is mentioned that references 17 and 22 provide a better description of the algorithm. In my opinion, the manuscript should be self-contained, and the algorithm should be better explained here.

In the caption of, for example, Fig. 2, it is mentioned that the "average computational entropy V(T)" is presented; however, in the main text, the manuscript states that the figure presents "non-repeatability". Again, this approach to presenting is quite confusing and does not help in appreciating the true value of this work.

Again, in Sec. 2, the letter R denotes "entries", and in the results, "records". Therefore, I believe a clear definition of R should be provided somewhere at the beginning.

In Fig. 3, both the regular mutability and sorted mutability exhibit a maximum at the critical temperature. If one zooms on the case of regular mutability, two well-defined peaks should be observed in contracts to what is stated in the manuscript.

On page 7, line 208, "T \approx 2.7" should be replaced by T \approx 2.27.

More than half of the references are from at least one of the authors of this manuscript. That's a bit too much for a short manuscript of this type. It gives the impression that the work is not novel enough or it is simply an incremental work.

In my opinion, this work can not be published in its current form. I also think that the calculations presented are an interesting exercise but do not necessarily need to be published.

Author Response

(The authors gave the same response as above.)

Reviewer 3 Report

Comments and Suggestions for Authors

This manuscript proposes an entropy function to quantify the mutability of complex systems. It is reasonable to use an entropic function to quantify the system's mutability as the change of the entropy for different systems can show the intrinsic property of it. However, the mathematical representation in this manuscript makes it very hard to follow. I hope the authors can change their description and make this paper easy to follow.

1.The definition of the Hamiltonian of the rising mode is weird. Why do the authors use S_jk and S_lm to represent different spins in this system? Please use the classical definition of the Hamiltonian of the 2D rising model. You can check it on Wikipedia.

2. There is the same problem for the definition of Dipolar Magnets. Especially, the definition of r jk,lm should be clearly defined. 

3. The definition of the map of M_w should be given clearly, as well as the definition of M_w itself. Nobody wants to check the important definition of a concept from the reference; the authors should give it in their manuscript.

4. The definitions of regular mutability and sorted mutability are also not clear.

If the authors clarify those definitions, this paper can show us interesting results.

Author Response

(The authors gave the same response as above.)

Round 2

Reviewer 1 Report

Comments and Suggestions for Authors

The authors have thoroughly addressed all of the concerns raised in my previous review. The revised manuscript is clear, technically sound, and well-structured. It presents a solid contribution to the field, and I now recommend it for publication in Entropy.

A minor suggestion:  In Eq. (1), replace “1.0” with “1” to reflect that the relation is exact, not approximate.

Reviewer 2 Report

Comments and Suggestions for Authors

The authors address my comments and concerns. In my opinion the new version improved. It can now be published in this journal.

Reviewer 3 Report

Comments and Suggestions for Authors

The revision has fixed my concern, and can be accepted for publication.